# Experimental Investigation of Phase Equilibria in the Co-Re-Ta Ternary System

**Xingjun Liu** [1,2]**, Dan Wu** [1]**, Jinbin Zhang** [1]**, Mujin Yang** [1]**, Jiahua Zhu** [1]**, Lingling Li** [1]**,
Yuechao Chen** [1]**, Shuiyuan Yang** [1]**, Jiajia Han** [1]  **, Yong Lu** [1] **and Cuiping Wang** [1,*]

[1] College of Materials and Fujian Provincial Key Laboratory of Materials Genome, Xiamen University, Xiamen 361005, China; lxj@xmu.edu.cn (X.L.); wudan225@163.com (D.W.); jbzhang@xmu.edu.cn (Jin.Z.); mujinguagua123@gmail.com (M.Y.); zhujiahua@stu.xmu.edu.cn (Jia.Z.); xmulee@foxmail.com (L.L.); cycofxmu@foxmail.com (Y.C.); yangshuiyuan@xmu.edu.cn (S.Y.); jiajiahan@xmu.edu.cn (J.H.); luyong@xmu.edu.cn (Y.L.)
[2] State Key Laboratory of Advanced Welding and Joining, Harbin Institute of Technology, Shenzhen 518005, China
[*] Correspondence: wangcp@xmu.edu.cn; Tel.: +86-592-218-0606; Fax: +86-592-218-7966

**Abstract:** In this study, the isothermal sections of the Co-Re-Ta ternary system at 1100, 1200, and 1300 °C have been experimentally investigated by means of electron probe microanalysis and X-ray diffraction. The results indicated the following: (1) The solid solubilities of the $\lambda_3$, ($\varepsilon$Co, Re), $\chi$-Re$_7$Ta$_3$, and bcc-(Ta) phases were large and changed very little from 1100 to 1300 °C; (2) more interestingly, the $\lambda_2$ phase, with a very limited solubility of Re, was surrounded by the $\lambda_3$ phase; (3) the solubility of Re for the $\mu$-Co$_6$Ta$_7$ phase increased slowly from 1100 to 1300 °C. These experimental results will be useful for Co-based high-temperature alloys, especially as a supplement for thermodynamic databases.

**Keywords:** Co-Re-Ta; phase diagram; high-temperature alloys; electron probe microanalysis

## 1. Introduction

It was first found by Sato et al. [1] that Co-based superalloys were strengthened by a stable ternary compound, $\gamma'$ Co$_3$(Al, W), with the L1$_2$ structure. Subsequently, the Co-based superalloys have been regarded as one of the promising high-temperature materials that exhibit better high-temperature strength than the conventional Ni-based superalloys [1]. To further improve high-temperature properties of Co-based superalloys, some refractory alloying elements, such as Re, Ta, W, and Mo, have been added to materials, which can improve high-temperature mechanical properties, creep properties, corrosion, and oxidation resistance [2–11]. Re does not randomly distribute in the alloy; it hinders dislocation movement by forming tiny clusters which act as obstacles during creep tests [2–5]. Thus, the addition of Re can effectively enhance the creep properties of superalloys. Meanwhile, as Re content increases, it refines the morphology and enhances the content of the alloy compound for Co-based superalloys [6]. Doping of Ta can maintain good microstructural stability and improve oxidation resistance [7,8]. The amount of Re and Ta additions is strictly restricted because excessive additions will cause the brittle and detrimental TCP (topologically close packed) phases to form at high stresses and temperatures [9–11]. Therefore, the knowledge of phase equilibria in the Co-Re-Ta ternary system is essential, which will provide significant basic data for the design of Co-based superalloys. However, there is no information on the experimental investigation of and thermodynamic data for the Co-Re-Ta ternary system. It is necessary to investigate the phase equilibria of the Co-Re-Ta ternary system.

The three binary systems, Co-Re, Re-Ta, and Co-Ta, that constitute the Co-Re-Ta ternary system are shown in Figure 1.

Elliott [12] has published the results for the Co-Re binary system. Later, Predel [13] reported the results for the Co-Re binary system based on experimental data. Recently, Liu et al. [14] and Guo et al. [15] estimated the Co-Re system and their findings were consistent with the experimental data. The newly assessed Co-Re phase diagram by Guo et al. [15] was applied in this work. The Co-Re system [15] is simple because there are two solid phases of ($\alpha$Co) and ($\varepsilon$Co, Re) and no intermediate phases. The ($\varepsilon$Co, Re) phase has a wide homogeneity range.

The Re-Ta binary system was first studied by Greenfield and Beck [16]. They investigated alloys with Ta contents between 25 and 52 at. % and reported the composition range of the $\sigma$ and $\chi$ phases. Cui and Jin [17] treated the $\sigma$ phase as a stoichiometric phase and thermodynamically assessed the Re-Ta system. Afterwards, Liu and Chang [18] also evaluated the thermodynamic description of the Re-Ta system. Recently, Guo et al. [15] estimated the Re-Ta phase diagram with the latest thermodynamic description for pure Re, and this Re-Ta binary system was used in the paper. The two intermediate phases of $\chi$-$Re_7Ta_3$ and $\sigma$-$Re_3Ta_2$ exist in the Re-Ta binary system [15]. The $\sigma$-$Re_3Ta_2$ phase forms from the peritectic reaction of liquid + $\chi$-$Re_7Ta_3$ ↔ $\sigma$-$Re_3Ta_2$. The melting point of the $\chi$-$Re_7Ta_3$ phase is about 2832 °C.

The Co-Ta binary system has been investigated by many researchers [19–24]. Itoh et al. [19] investigated the homogeneity ranges, crystal structures, and magnetic properties of the three Laves phases in the Co-Ta system. Okamoto [20] assessed the Co-Ta phase diagram and treated C14 as a line compound. Liu and Chang [21] thermodynamically assessed the Co-Ta binary system. Shinagawa et al. [24] studied the Co-Ta binary system and revised the $\lambda_1$ phase as an intermetallic compound with a narrow composition range, and the phase diagram [24] was adopted in this work. There are three Laves phases of $\lambda_1$ (C14), $\lambda_2$ (C15), and $\lambda_3$ (C36), each with different polytypes. The other intermediate phases of $CoTa_2$, $\mu$-$Co_6Ta_7$, and $Co_7Ta_2$, and solution phases of bcc-(Ta), ($\varepsilon$Co), and ($\alpha$Co) also exist in the Co-Ta binary system. The $\lambda_1$ and $\lambda_3$ phases are obtained from peritectic reactions of liquid + $\lambda_2$ ↔ $\lambda_1$ and liquid + $\lambda_2$ ↔ $\lambda_3$, respectively. The $\lambda_1$ phase exists at ~1293–1587 °C and the $\lambda_3$ phase exists at ~947–1456 °C. The $Co_7Ta_2$ phase with the $BaPb_3$ [23] crystal structure is obtained from a peritectoid reaction: ($\alpha$Co) + $\lambda_3$ ↔ $Co_7Ta_2$. The information for the stable solid phases and their crystal structures in the three binary systems is listed in Table 1.

The temperatures of 1100, 1200, and 1300 °C were selected because the Co-based superalloys are widely used in high-temperature areas such as aircraft engines and turbine blades. Thus, it is more meaningful to investigate the phase equilibria at high temperatures. The present work aimed to experimentally investigate the phase equilibria of the Co-Re-Ta ternary system at 1100, 1200, and 1300 °C using electron probe microanalysis and X-ray diffraction techniques in order to understand the microstructures of the Co-Re-Ta ternary system and provide useful information for the development of Co-based high-temperature alloys.

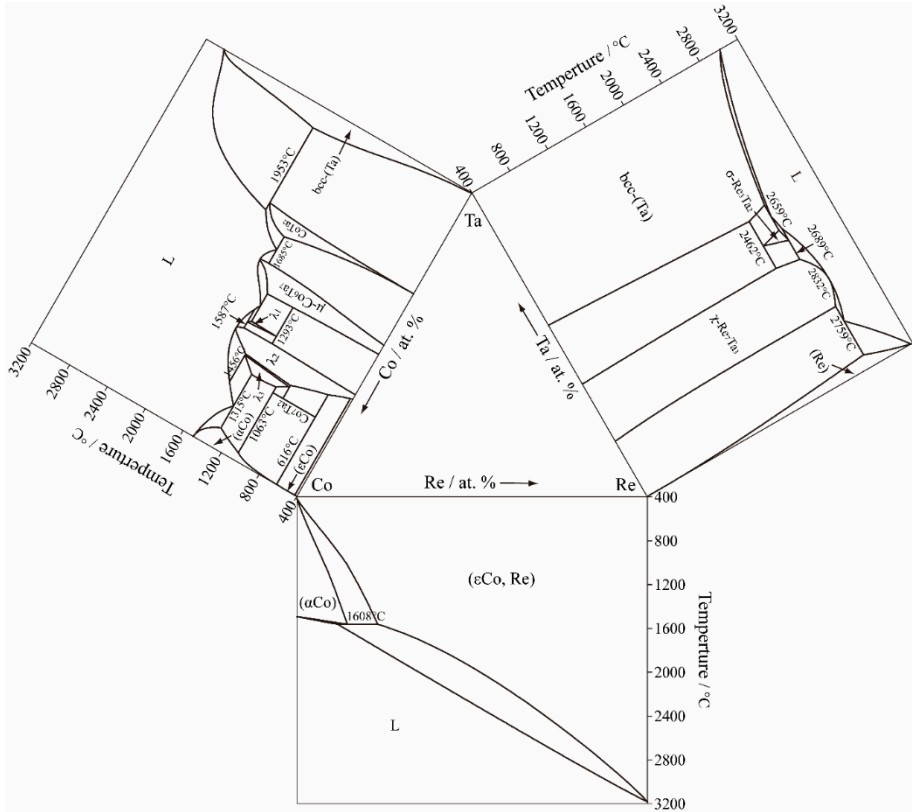

**Figure 1.** Binary phase diagrams constituting the Co-Re-Ta ternary system [15,24].

**Table 1.** Crystal structures of each phase in the Co-Re-Ta ternary system [15,23,24].

| System | Phase | Pearson Symbol | Space Group | Prototype | Structure Type | References |
|--------|-------|----------------|-------------|-----------|----------------|------------|
| Re-Ta | (Re) | hP2 | P6₃/mmc | Mg | A3 | [15] |
| | χ-Re₇Ta₃ | cI58 | I-43m | αMn | A12 | [15] |
| | σ-Re₃Ta₂ | tP30 | P4₂/mnm | σCrFe | D8ᵦ | [15] |
| | bcc-(Ta) | cI2 | Im-3m | W | A2 | [15] |
| Co-Re | (αCo) | cF4 | Fm-3m | Cu | A1 | [15] |
| | (εCo, Re) | hP2 | P6₃/mmc | Mg | A3 | [15] |
| Co-Ta | (αCo) | cF4 | Fm-3m | Cu | A1 | [24] |
| | (εCo) | hP2 | P6₃/mmc | Mg | A3 | [24] |
| | λ₃-Co₂Ta | hP24 | P6₃/mmc | Ni₂Mg | C36 | [24] |
| | λ₂-Co₂Ta | cF24 | Fd-3m | Cu₂Mg | C15 | [24] |
| | λ₁-Co₂Ta | hP12 | P6₃/mmc | Zn₂Mg | C14 | [24] |
| | μ-Co₆Ta₇ | hR13 | R-3m | Fe₇W₆ | D8ᵦ | [24] |
| | CoTa₂ | tI12 | I4/mcm | Al₂Cu | C16 | [24] |
| | bcc-(Ta) | cI2 | Im-3m | W | A2 | [24] |
| | Co₇Ta₂ | hR36 | R-3m | BaPb₃ [23] | - | [23,24] |

## 2. Experimental Procedure

High-purity rhenium (99.9 wt %), tantalum (99.9 wt %), and cobalt (99.9 wt %) were used as raw materials. The required weights of the elements (with a total weight of about 20 g) were measured with a semi-micro analytical balance with an accuracy of at least 0.5 mg. During the whole sample preparation procedure, the mass loss was usually less than 1%. Therefore, the mass loss was assumed to make no significant effect on the sample composition.

All the bulk alloys were prepared in the form of atomic ratios (at. %). The bulk alloys with nominal compositions were melted by arc-melting in an argon atmosphere using a non-consumable tungsten electrode on a water-cooled plate. Titanium was used as a getter material. The buttons were re-melted at least five times to ensure that the ingots were homogeneous.

Afterwards, the specimens were cut into small pieces by a wire-cutting machine for heat treatment. The samples were cleaned by an ultrasonic cleaner and then encapsulated in quartz ampoules which were evacuated and flushed several times with purified argon. Heat treatments were performed at 1100, 1200, and 1300 °C, respectively. The time of the heat treatments ranged from 15 days to 65 days to reach phase equilibria based on the different temperatures and compositions of the samples. The samples containing over 20 at. % Re were heat-treated for a relatively long time. Subsequently, the specimens were quenched, mounted, grinded, and polished.

The microstructural observation and equilibrium composition analysis of specimens was characterized by electron probe microanalysis (EPMA) (JXA-8100R, JEOL, Tokyo, Japan). Pure elements were used as standards and the measurements were carried out at a voltage of 20 kV and a current of $1.0 \times 10^{-8}$ A. To identify the crystal structures, powder X-ray diffraction (XRD) measurements were performed on a Philips Panalytical X-pert diffractometer (Bruker Daltonic Inc., Billerica, MA, USA) with Cu K$\alpha$ radiation at 40 kV and 40 mA. The scanning range of 2θ was from 20° to 90° at a step size of 0.0167°.

## 3. Results and Discussion

### 3.1. Microstructure

The typical back-scattered electron (BSE) images of ternary Co-Re-Ta alloys annealed at 1100, 1200, or 1300 °C for different times are shown in Figure 2; there are three-phase equilibrium microstructures shown in Figure 2a–d and two-phase equilibrium microstructures shown in Figure 2e–j. The corresponding results of the XRD are presented in Figure 3.

For the $Co_{74}Re_{16}Ta_{10}$ alloy, the light grey phase (($\varepsilon$Co, Re)), dark grey phase ($\lambda_3$), and black phase (($\alpha$Co)) were observed after annealing at 1100 °C for 50 days, as shown in Figure 2a. Figure 2b shows the three-phase equilibrium of the $\mu$-$Co_6Ta_7$ phase, $\lambda_3$ phase, and bcc-(Ta) phase in the 1200 °C/35 days-annealed $Co_{44}Re_{14}Ta_{42}$ alloy. The white phase was bcc-(Ta), the light grey phase was $\mu$-$Co_6Ta_7$, and the dark grey phase was $\lambda_3$. Figure 3a shows that the corresponding XRD pattern, and the $\mu$-$Co_6Ta_7$ phase, $\lambda_3$ phase, and bcc-(Ta) phase were clearly distinguished by the different symbols. The $Co_{31}Re_{35}Ta_{34}$ alloy annealed at 1300 °C for 25 days contained the three phases of $\chi$-$Re_7Ta_3$ (light grey), bcc-(Ta) (dark grey), and $\lambda_3$ (black), which are shown in Figure 2c. The $\lambda_3$ phase was the matrix while the bcc-(Ta) phase was on the edge of the $\chi$-$Re_7Ta_3$ phase. XRD identification, as shown in Figure 3b, confirmed the existence of the three phases of $\chi$-$Re_7Ta_3$, bcc-(Ta), and $\lambda_3$. Figure 2d shows the BSE image of the $Co_{45}Re_{30}Ta_{25}$ alloy annealed at 1300 °C for 25 days. There were three phases of $\lambda_3$, $\chi$-$Re_7Ta_3$, and ($\varepsilon$Co, Re) existing in an equilibrium. Figure 2e shows a two-phase microstructure constituted by a white ($\varepsilon$Co, Re) phase and a black $\lambda_3$ phase in the $Co_{54}Re_{25}Ta_{21}$ alloy quenched from 1100 °C. The two-phase equilibrium of bcc-(Ta) and $CoTa_2$ was identified in the annealed $Co_{22}Re_5Ta_{73}$ alloy (1100 °C/50 days), as shown in Figure 2f. In the $Co_{58}Re_3Ta_{39}$ alloy annealed at 1200 °C for 35 days, the white $\mu$-$Co_6Ta_7$ phase and the dark grey $\lambda_3$ phase were observed in Figure 2g while their crystal structure was confirmed by the XRD pattern in Figure 3c. Figure 2h shows the BSE image of the bcc-(Ta) phase and the $\mu$-$Co_6Ta_7$ phase in the $Co_{25}Re_{20}Ta_{55}$ alloy annealed at 1200 °C for 50 days. The white bcc-(Ta) phase was homogeneously distributed in the grey $\mu$-$Co_6Ta_7$ phase. Figure 2i shows that the two phases of $\lambda_3$ and ($\alpha$Co) were found in the $Co_{80}Re_{11}Ta_9$ alloy annealed at 1300 °C for 15 days. The light grey phase was $\lambda_3$ and the dark grey phase was ($\alpha$Co). The two-phase microstructure of the $\lambda_2$ phase (dark grey) and $\mu$-$Co_6Ta_7$ phase (light grey) was identified in the $Co_{60}Re_1Ta_{39}$ alloy annealed at 1300 °C for 15 days, as shown in Figure 2j. The corresponding XRD pattern is displayed in Figure 3d.

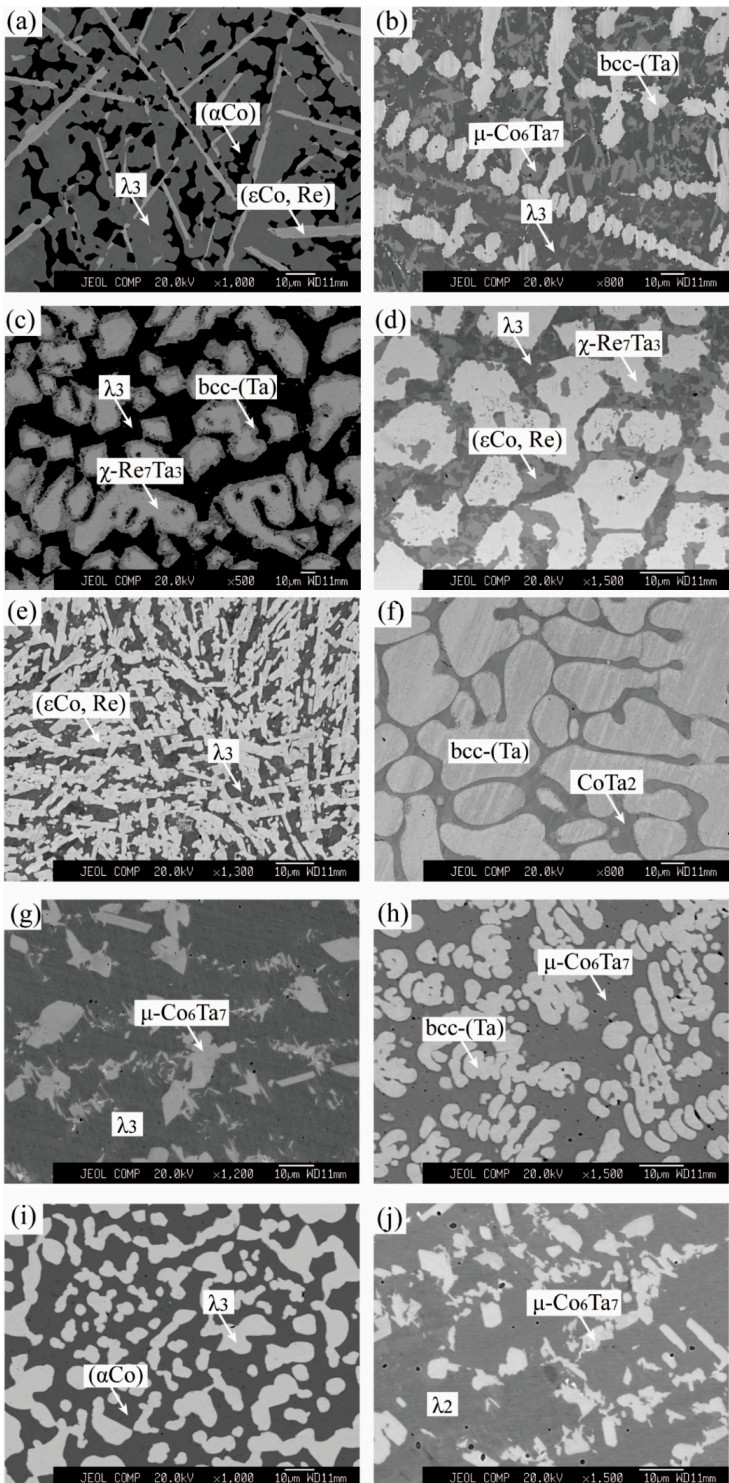

**Figure 2.** Back-scattered electron (BSE) images of the typical ternary Co-Re-Ta alloys: (**a**) The Co$_{74}$Re$_{16}$Ta$_{10}$ alloy annealed at 1100 °C for 50 days; (**b**) the Co$_{44}$Re$_{14}$Ta$_{42}$ alloy annealed at 1200 °C for 35 days; (**c**) the Co$_{31}$Re$_{35}$Ta$_{34}$ alloy annealed at 1300 °C for 25 days; (**d**) the Co$_{45}$Re$_{30}$Ta$_{25}$ alloy annealed at 1300 °C for 25 days; (**e**) the Co$_{54}$Re$_{25}$Ta$_{21}$ alloy annealed at 1100 °C for 65 days; (**f**) the Co$_{22}$Re$_5$Ta$_{73}$ alloy annealed at 1100 °C for 50 days; (**g**) the Co$_{58}$Re$_3$Ta$_{39}$ alloy annealed at 1200 °C for 35 days; (**h**) the Co$_{25}$Re$_{20}$Ta$_{55}$ alloy annealed at 1200 °C for 50 days; (**i**) the Co$_{80}$Re$_{11}$Ta$_9$ alloy annealed at 1300 °C for 15 days; (**j**) the Co$_{60}$Re$_1$Ta$_{39}$ alloy annealed at 1300 °C for 15 days.

In order to figure out the phase boundary between the $\lambda_3$ and $\lambda_2$ phases, several alloys were prepared. Unfortunately, all the compositions were located at single field region, which meant that the $\lambda_2/\lambda_3$ two-phase field was extremely narrow. This was consistent with that of the Co-Ta binary. The phase boundaries were then plotted with approximations based on the microstructure observation results of these single-phased compositions. Figure 4a,b shows the typical XRD patterns of the $Co_{66}Re_5Ta_{29}$ and $Co_{68}Re_3Ta_{29}$ alloys annealed at 1300 °C for 15 days, which were confirmed to be $\lambda_3$ and $\lambda_2$ single phases, respectively. The corresponding microstructure was consistent with the results.

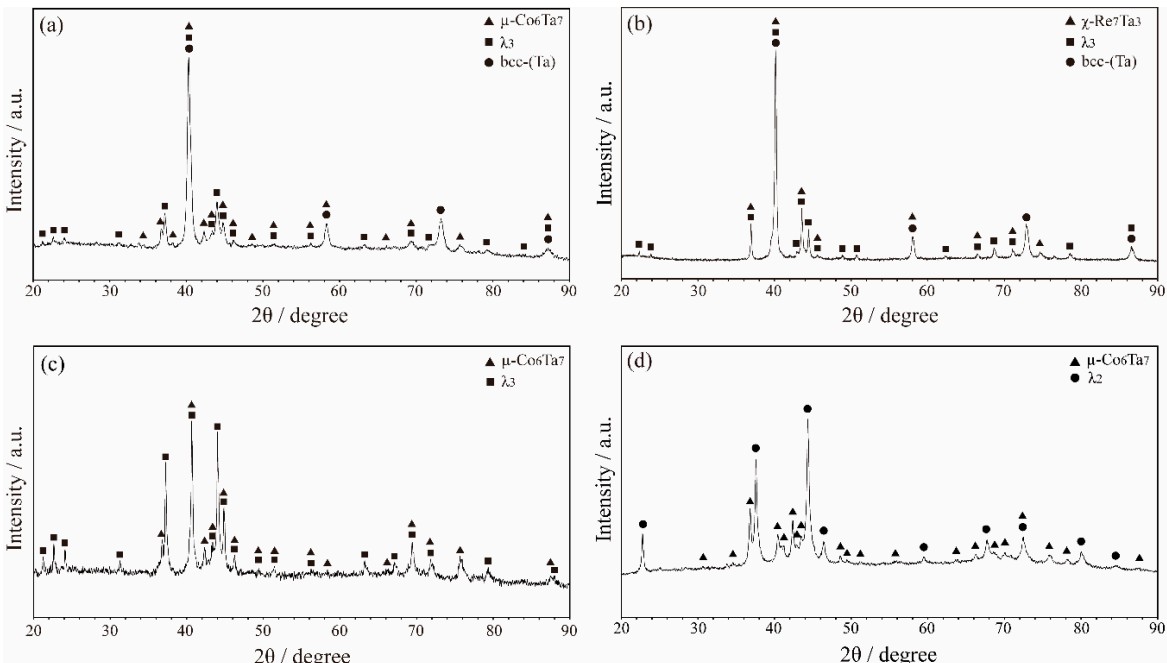

**Figure 3.** X-ray diffraction patterns obtained from (**a**) the $Co_{44}Re_{14}Ta_{42}$ alloy annealed at 1200 °C for 35 days, (**b**) the $Co_{31}Re_{35}Ta_{34}$ alloy annealed at 1300 °C for 25 days, (**c**) the $Co_{58}Re_3Ta_{39}$ alloy annealed at 1200 °C for 35 days, and (**d**) the $Co_{60}Re_1Ta_{39}$ alloy annealed at 1300 °C for 15 days.

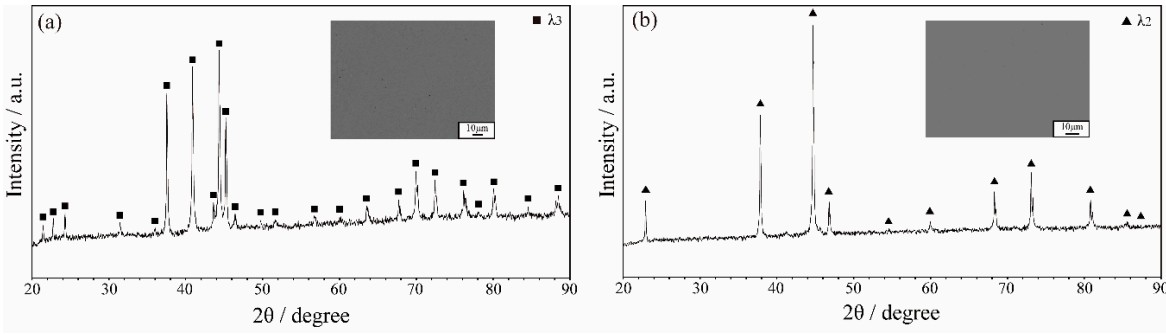

**Figure 4.** X-ray diffraction patterns obtained from (**a**) the $Co_{66}Re_5Ta_{29}$ alloy annealed at 1200 °C for 35 days and (**b**) the $Co_{68}Re_3Ta_{29}$ alloy annealed at 1300 °C for 15 days.

### 3.2. Isothermal Sections

All the equilibrium compositions of the Co-Re-Ta ternary system at 1100, 1200, and 1300 °C are listed in Tables 2–4, respectively. Figure 5a–c shows the isothermal sections at 1100, 1200, and 1300 °C based on the experimental data, respectively. The $\lambda_2$ single phase, $\lambda_3$ single phase, two-phase equilibrium, and three-phase equilibrium are characterized by different symbols. The solid triangle represents the determined three-phase equilibrium while the dashed triangle represents the undetermined three-phase equilibrium.

**Table 2.** Equilibrium compositions of the Co-Re-Ta ternary system at 1100 °C as determined in the present work.

| Nominal Alloys (at. %) | Annealed Time (days) | Phase Equilibrium Phase 1/Phase 2/Phase 3 | Composition (at. %) | | | | | |
|---|---|---|---|---|---|---|---|---|
| | | | Phase 1 | | Phase 2 | | Phase 3 | |
| | | | Re | Ta | Re | Ta | Re | Ta |
| $Co_{31}Re_{35}Ta_{34}$ | 65 | $\lambda_3$/bcc-(Ta)/$\chi$-$Re_7Ta_3$ | 19.7 | 26.5 | 46.5 | 45.5 | 52.7 | 34.8 |
| $Co_{45}Re_{30}Ta_{25}$ | 65 | $\lambda_3$/($\varepsilon$Co, Re)/$\chi$-$Re_7Ta_3$ | 20.1 | 25.5 | 30.4 | 22.0 | 60.1 | 26.0 |
| $Co_{60}Re_{20}Ta_{20}$ | 65 | ($\varepsilon$Co, Re)/$\lambda_3$ | 27.3 | 16.6 | 16.5 | 20.5 | | |
| $Co_{45}Re_{20}Ta_{35}$ | 65 | bcc-(Ta)/$\lambda_3$ | 41.4 | 48.4 | 13.1 | 30.9 | | |
| $Co_{25}Re_{20}Ta_{55}$ | 65 | $\mu$-$Co_6Ta_7$/bcc-(Ta) | 2.3 | 50.6 | 34.1 | 58.3 | | |
| $Co_{54}Re_{25}Ta_{21}$ | 65 | ($\varepsilon$Co, Re)/$\lambda_3$ | 28.7 | 19.1 | 18.4 | 22.9 | | |
| $Co_{78}Re_5Ta_{17}$ | 50 | ($\alpha$Co)/$\lambda_3$ | 2.7 | 3.3 | 4.7 | 21.8 | | |
| $Co_{44}Re_{14}Ta_{42}$ | 50 | $\lambda_3$/$\mu$-$Co_6Ta_7$/bcc-(Ta) | 2.7 | 33.8 | 3.8 | 43.7 | 39.2 | 51.2 |
| $Co_{22}Re_5Ta_{73}$ | 50 | $CoTa_2$/bcc-(Ta) | 0.7 | 65.8 | 8.9 | 87.5 | | |
| $Co_{58}Re_3Ta_{39}$ | 50 | $\lambda_3$/$\mu$-$Co_6Ta_7$ | 1.9 | 33.6 | 3.2 | 43.4 | | |
| $Co_{80}Re_2Ta_{18}$ | 50 | ($\alpha$Co)/$\lambda_3$ | 0.9 | 3.5 | 1.8 | 22.7 | | |
| $Co_{80}Re_5Ta_{15}$ | 50 | ($\alpha$Co)/$\lambda_3$ | 3.5 | 3.6 | 5.7 | 20.8 | | |
| $Co_{80}Re_{11}Ta_9$ | 50 | ($\alpha$Co)/$\lambda_3$ | 9.2 | 2.7 | 13.8 | 17.2 | | |
| $Co_{74}Re_{16}Ta_{10}$ | 50 | ($\alpha$Co)/$\lambda_3$/($\varepsilon$Co, Re) | 10.1 | 2.4 | 15.2 | 17.3 | 25.0 | 14.2 |
| $Co_{54}Re_{18}Ta_{28}$ | 50 | $\lambda_3$ | 18.2 | 28.3 | | | | |
| $Co_{89}Re_5Ta_6$ | 50 | ($\alpha$Co)/$\lambda_3$ | 4.6 | 3.7 | 7.5 | 19.6 | | |
| $Co_{33}Re_5Ta_{62}$ | 50 | $CoTa_2$/bcc-(Ta) | 1.7 | 59.2 | 18.0 | 77.3 | | |
| $Co_{30}Re_{10}Ta_{60}$ | 50 | $\mu$-$Co_6Ta_7$/bcc-(Ta) | 1.7 | 54.2 | 24.0 | 71.3 | | |
| $Co_{69}Re_9Ta_{22}$ | 50 | $\lambda_3$ | 9.3 | 22.3 | | | | |
| $Co_{70}Re_5Ta_{25}$ | 50 | $\lambda_3$ | 4.5 | 24.9 | | | | |
| $Co_{71}Re_1Ta_{28}$ | 50 | $\lambda_2$ | 0.8 | 28.2 | | | | |
| $Co_{68}Re_3Ta_{29}$ | 50 | $\lambda_2$ | 2.7 | 28.7 | | | | |
| $Co_{66}Re_5Ta_{29}$ | 50 | $\lambda_3$ | 4.7 | 28.5 | | | | |
| $Co_{64}Re_7Ta_{29}$ | 50 | $\lambda_3$ | 7.3 | 28.9 | | | | |
| $Co_{60}Re_1Ta_{39}$ | 50 | $\lambda_2$/$\mu$-$Co_6Ta_7$ | 0.7 | 32.3 | 1.2 | 43.7 | | |
| $Co_{28}Re_6Ta_{66}$ | 50 | $CoTa_2$/bcc-(Ta) | 1.4 | 60.2 | 14.2 | 80.7 | | |
| $Co_{43}Re_1Ta_{56}$ | 50 | $CoTa_2$/$\mu$-$Co_6Ta_7$ | 0.9 | 57.8 | 1.1 | 54.8 | | |

**Table 3.** Equilibrium compositions of the Co-Re-Ta ternary system at 1200 °C as determined in the present work.

| Nominal Alloys (at. %) | Annealed Time (days) | Phase Equilibrium Phase 1/Phase 2/Phase 3 | Composition (at. %) | | | | | |
|---|---|---|---|---|---|---|---|---|
| | | | Phase 1 | | Phase 2 | | Phase 3 | |
| | | | Re | Ta | Re | Ta | Re | Ta |
| $Co_{31}Re_{35}Ta_{34}$ | 50 | $\lambda_3$/bcc-(Ta)/$\chi$-$Re_7Ta_3$ | 21.7 | 26.1 | 50.0 | 47.6 | 56.6 | 31.4 |
| $Co_{45}Re_{30}Ta_{25}$ | 50 | $\lambda_3$/($\varepsilon$Co, Re)/$\chi$-$Re_7Ta_3$ | 21.7 | 25.4 | 31.3 | 24.3 | 59.9 | 26.8 |
| $Co_{60}Re_{20}Ta_{20}$ | 50 | ($\varepsilon$Co, Re)/$\lambda_3$ | 27.2 | 18.7 | 17.2 | 20.9 | | |
| $Co_{45}Re_{20}Ta_{35}$ | 50 | bcc-(Ta)/$\lambda_3$ | 41.1 | 48.6 | 14.0 | 31.9 | | |
| $Co_{25}Re_{20}Ta_{55}$ | 50 | $\mu$-$Co_6Ta_7$/bcc-(Ta) | 2.8 | 51.1 | 35.3 | 58.7 | | |
| $Co_{54}Re_{25}Ta_{21}$ | 50 | ($\varepsilon$Co, Re)/$\lambda_3$ | 28.0 | 20.9 | 20.1 | 22.9 | | |
| $Co_{78}Re_5Ta_{17}$ | 35 | ($\alpha$Co)/$\lambda_3$ | 2.7 | 4.4 | 4.3 | 21.9 | | |
| $Co_{44}Re_{14}Ta_{42}$ | 35 | $\lambda_3$/$\mu$-$Co_6Ta_7$/bcc-(Ta) | 3.5 | 35.3 | 4.0 | 44.2 | 39.4 | 50.4 |
| $Co_{22}Re_5Ta_{73}$ | 35 | $CoTa_2$/bcc-(Ta) | 0.8 | 65.2 | 8.9 | 86.5 | | |
| $Co_{58}Re_3Ta_{39}$ | 35 | $\lambda_3$/$\mu$-$Co_6Ta_7$ | 2.4 | 34.4 | 3.3 | 43.9 | | |
| $Co_{80}Re_2Ta_{18}$ | 35 | ($\alpha$Co)/$\lambda_3$ | 1.4 | 4.9 | 1.8 | 22.5 | | |
| $Co_{80}Re_5Ta_{15}$ | 35 | ($\alpha$Co)/$\lambda_3$ | 3.2 | 4.2 | 5.5 | 21.1 | | |
| $Co_{80}Re_{11}Ta_9$ | 35 | ($\alpha$Co)/$\lambda_3$ | 9.7 | 3.0 | 13.6 | 18.4 | | |
| $Co_{74}Re_{16}Ta_{10}$ | 35 | ($\alpha$Co)/$\lambda_3$/($\varepsilon$Co, Re) | 11.2 | 2.8 | 15.5 | 17.6 | 24.9 | 15.3 |
| $Co_{54}Re_{18}Ta_{28}$ | 35 | $\lambda_3$ | 18.3 | 28.2 | | | | |
| $Co_{89}Re_5Ta_6$ | 35 | ($\alpha$Co)/$\lambda_3$ | 4.3 | 4.5 | 7.1 | 20.6 | | |
| $Co_{33}Re_5Ta_{62}$ | 35 | $CoTa_2$/bcc-(Ta) | 1.5 | 59.6 | 19.5 | 76.8 | | |
| $Co_{30}Re_{10}Ta_{60}$ | 35 | $\mu$-$Co_6Ta_7$/bcc-(Ta) | 2.0 | 55.1 | 23.5 | 72.2 | | |
| $Co_{69}Re_9Ta_{22}$ | 35 | $\lambda_3$ | 8.4 | 22.0 | | | | |
| $Co_{70}Re_5Ta_{25}$ | 35 | $\lambda_3$ | 4.5 | 24.8 | | | | |
| $Co_{71}Re_1Ta_{28}$ | 35 | $\lambda_2$ | 0.8 | 28.0 | | | | |
| $Co_{68}Re_3Ta_{29}$ | 35 | $\lambda_2$ | 2.6 | 28.8 | | | | |
| $Co_{66}Re_5Ta_{29}$ | 35 | $\lambda_3$ | 4.8 | 29.0 | | | | |
| $Co_{64}Re_7Ta_{29}$ | 35 | $\lambda_3$ | 7.0 | 28.8 | | | | |
| $Co_{60}Re_1Ta_{39}$ | 35 | $\lambda_2$/$\mu$-$Co_6Ta_7$ | 0.5 | 32.4 | 1.0 | 44.1 | | |
| $Co_{28}Re_6Ta_{66}$ | 35 | $CoTa_2$/bcc-(Ta) | 1.3 | 60.2 | 15.9 | 79.6 | | |
| $Co_{43}Re_1Ta_{56}$ | 35 | $CoTa_2$/$\mu$-$Co_6Ta_7$ | 0.6 | 60.5 | 1.3 | 54.8 | | |

**Table 4.** Equilibrium compositions of the Co-Re-Ta ternary system at 1300 °C as determined in the present work.

| Nominal Alloys (at. %) | Annealed Time (days) | Phase Equilibrium Phase 1/Phase 2/Phase 3 | Composition (at. %) | | | | | |
|---|---|---|---|---|---|---|---|---|
| | | | Phase 1 | | Phase 2 | | Phase 3 | |
| | | | Re | Ta | Re | Ta | Re | Ta |
| $Co_{31}Re_{35}Ta_{34}$ | 25 | $\lambda_3$/bcc-(Ta)/χ-Re$_7$Ta$_3$ | 22.2 | 27.1 | 49.0 | 48.5 | 56.6 | 32.6 |
| $Co_{45}Re_{30}Ta_{25}$ | 25 | $\lambda_3$/(εCo, Re)/χ-Re$_7$Ta$_3$ | 22.6 | 26.1 | 32.0 | 23.2 | 58.7 | 25.8 |
| $Co_{60}Re_{20}Ta_{20}$ | 25 | (εCo, Re)/$\lambda_3$ | 27.5 | 17.0 | 17.3 | 20.6 | | |
| $Co_{45}Re_{20}Ta_{35}$ | 25 | bcc-(Ta)/$\lambda_3$ | 42.8 | 48.9 | 14.2 | 32.8 | | |
| $Co_{25}Re_{20}Ta_{55}$ | 25 | μ-Co$_6$Ta$_7$/bcc-(Ta) | 3.9 | 51.2 | 35.5 | 58.9 | | |
| $Co_{54}Re_{25}Ta_{21}$ | 25 | (εCo, Re)/$\lambda_3$ | 28.8 | 18.9 | 19.7 | 22.1 | | |
| $Co_{78}Re_5Ta_{17}$ | 15 | (αCo)/$\lambda_3$ | 2.8 | 5.5 | 5.0 | 21.3 | | |
| $Co_{44}Re_{14}Ta_{42}$ | 15 | $\lambda_3$/μ-Co$_6$Ta$_7$/bcc-(Ta) | 5.5 | 36.4 | 6.1 | 43.2 | 41.8 | 50.4 |
| $Co_{22}Re_5Ta_{73}$ | 15 | CoTa$_2$/bcc-(Ta) | 0.8 | 65.2 | 8.7 | 87.6 | | |
| $Co_{58}Re_3Ta_{39}$ | 15 | $\lambda_3$/μ-Co$_6$Ta$_7$ | 2.7 | 35.4 | 3.6 | 44.0 | | |
| $Co_{80}Re_2Ta_{18}$ | 15 | (αCo)/$\lambda_3$ | 1.3 | 5.6 | 1.9 | 22.3 | | |
| $Co_{80}Re_5Ta_{15}$ | 15 | (αCo)/$\lambda_3$ | 3.5 | 5.0 | 5.9 | 20.8 | | |
| $Co_{80}Re_{11}Ta_9$ | 15 | (αCo)/$\lambda_3$ | 10.0 | 3.8 | 13.8 | 17.8 | | |
| $Co_{74}Re_{16}Ta_{10}$ | 15 | (αCo)/$\lambda_3$/(εCo, Re) | 13.0 | 2.6 | 16.2 | 17.8 | 26.0 | 14.2 |
| $Co_{54}Re_{18}Ta_{28}$ | 15 | $\lambda_3$ | 18.2 | 28.5 | | | | |
| $Co_{89}Re_5Ta_6$ | 15 | (αCo)/$\lambda_3$ | 4.8 | 5.0 | 7.2 | 22.4 | | |
| $Co_{33}Re_5Ta_{62}$ | 15 | CoTa$_2$/bcc-(Ta) | 2.1 | 59.2 | 19.2 | 76.2 | | |
| $Co_{30}Re_{10}Ta_{60}$ | 15 | μ-Co$_6$Ta$_7$/bcc-(Ta) | 2.4 | 54.1 | 24.2 | 71.4 | | |
| $Co_{69}Re_9Ta_{22}$ | 15 | $\lambda_3$ | 8.9 | 22.0 | | | | |
| $Co_{70}Re_5Ta_{25}$ | 15 | $\lambda_3$ | 4.5 | 24.5 | | | | |
| $Co_{71}Re_1Ta_{28}$ | 15 | $\lambda_2$ | 0.7 | 27.8 | | | | |
| $Co_{68}Re_3Ta_{29}$ | 15 | $\lambda_2$ | 2.7 | 28.6 | | | | |
| $Co_{66}Re_5Ta_{29}$ | 15 | $\lambda_3$ | 4.6 | 28.4 | | | | |
| $Co_{64}Re_7Ta_{29}$ | 15 | $\lambda_3$ | 7.0 | 28.6 | | | | |
| $Co_{60}Re_1Ta_{39}$ | 15 | $\lambda_2$/μ-Co$_6$Ta$_7$ | 0.8 | 32.3 | 1.3 | 44.2 | | |
| $Co_{28}Re_6Ta_{66}$ | 15 | CoTa$_2$/bcc-(Ta) | 1.8 | 58.3 | 14.9 | 81.1 | | |
| $Co_{43}Re_1Ta_{56}$ | 15 | CoTa$_2$/μ-Co$_6$Ta$_7$ | 0.6 | 59.7 | 1.3 | 54.9 | | |

Figure 5a shows the 1100 °C isothermal section of the Co-Re-Ta ternary system. There were three solid solution phases of (αCo), (εCo, Re), and bcc-(Ta), two Laves phases of $\lambda_2$ and $\lambda_3$, and the intermetallic compounds of the μ-Co$_6$Ta$_7$, CoTa$_2$, and χ-Re$_7$Ta$_3$ phases. Investigations of the $Co_{74}Re_{16}Ta_{10}$, $Co_{44}Re_{14}Ta_{42}$, $Co_{31}Re_{35}Ta_{34}$, and $Co_{45}Re_{30}Ta_{25}$ alloys were used to determine the three-phase equilibria of (αCo) + (εCo, Re) + $\lambda_3$, $\lambda_3$ + bcc-(Ta) + μ-Co$_6$Ta$_7$, $\lambda_3$ + χ-Re$_7$Ta$_3$ + bcc-(Ta), and (εCo, Re) + $\lambda_3$ + χ-Re$_7$Ta$_3$, respectively. Seven alloys were confirmed to be single phases, two alloys ($Co_{71}Re_1Ta_{28}$ and $Co_{68}Re_3Ta_{29}$) were located in the $\lambda_2$ single-phase region, and five alloys ($Co_{54}Re_{18}Ta_{28}$, $Co_{69}Re_9Ta_{22}$, $Co_{70}Re_5Ta_{25}$, $Co_{66}Re_5Ta_{29}$ and $Co_{64}Re_7Ta_{29}$) were located in the $\lambda_3$ single-phase region. The solubility of Re in the $\lambda_3$ phase was measured to be about 20.1 at. % while the solubility of Re in the $\lambda_2$ phase was measured to be about 3.8 at. %. The $\lambda_3$ phase extended from the left side to the right side of the $\lambda_2$ phase and was wrapped around the $\lambda_2$ phase. The solubility of Re in the μ-Co$_6$Ta$_7$ and CoTa$_2$ phases was quite small at roughly 3.3 at. % and 1.5 at. %, respectively. The solubility of Co in the χ-Re$_7$Ta$_3$ phase was about 16.8 at. %. The (εCo, Re) phase extended from the Re-rich side to the Co-rich side, and the solubility of Ta in (εCo, Re) phase was about 21.7 at. %.

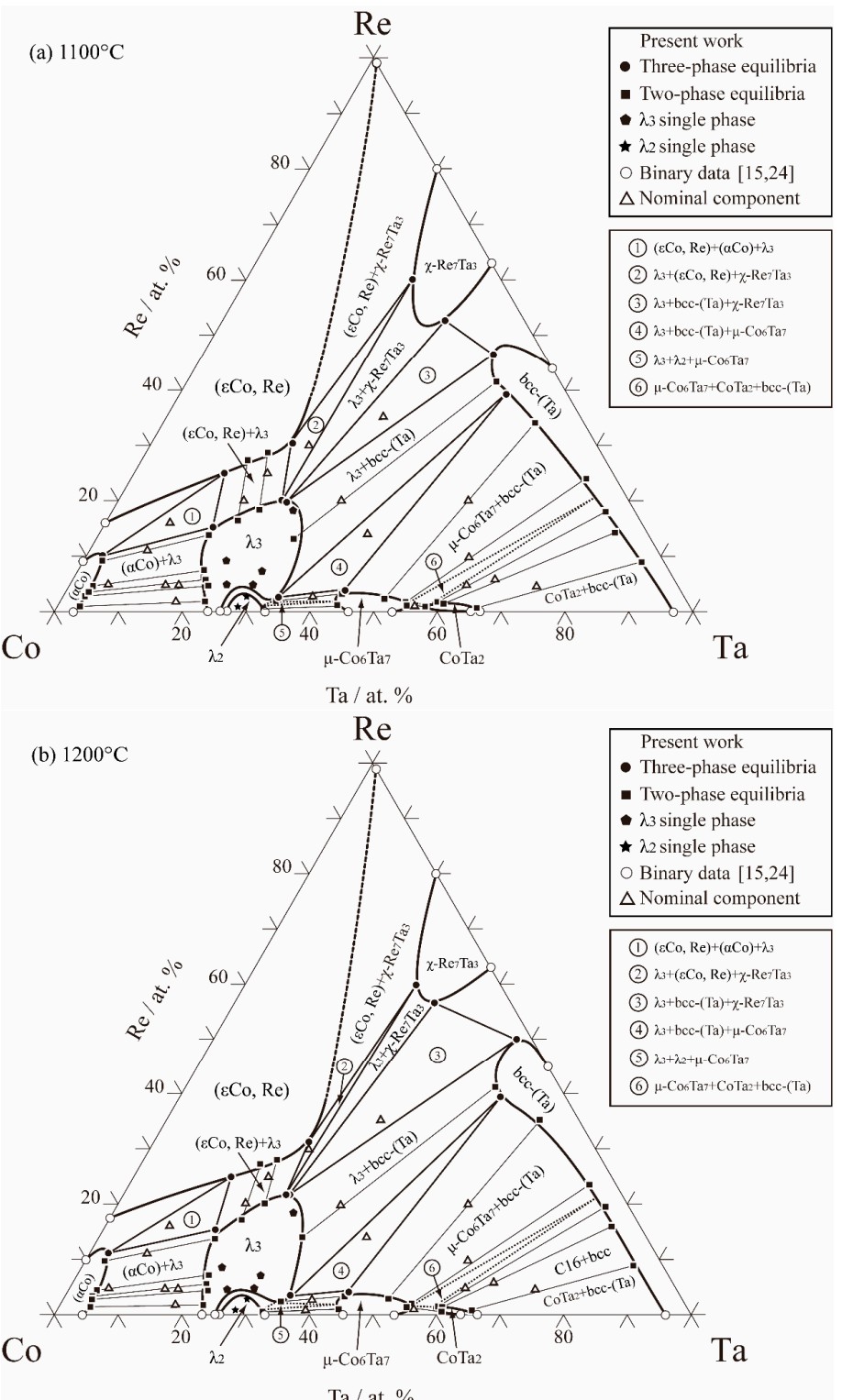

**Figure 5.** *Cont.*

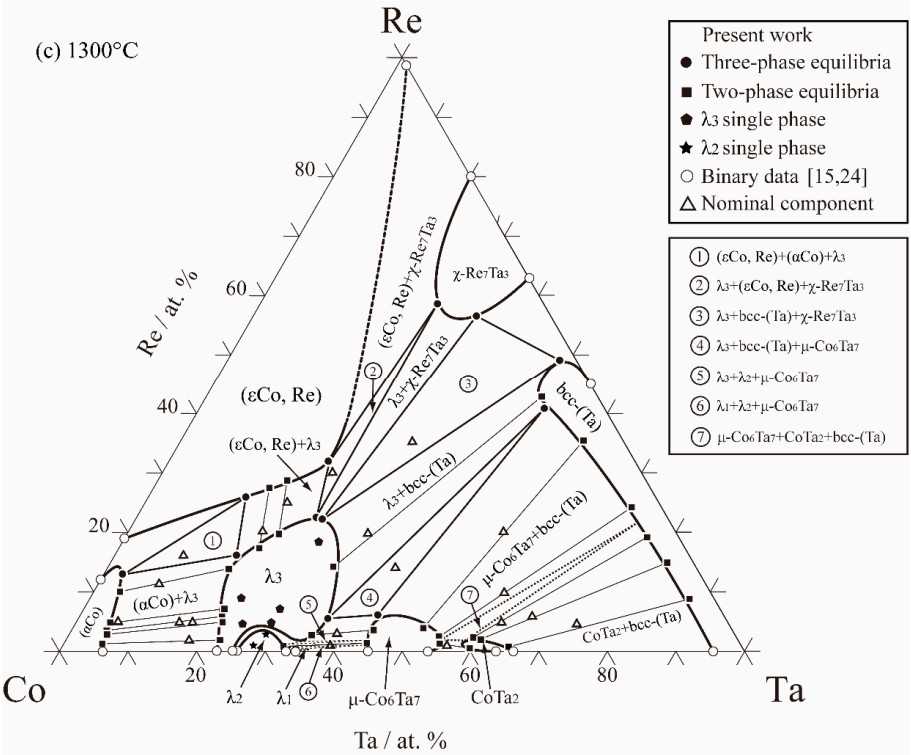

**Figure 5.** Experimentally determined isothermal section of the Co-Re-Ta system: (**a**) 1100 °C, (**b**) 1200 °C, (**c**) 1300 °C.

Figure 5b shows the isothermal section at 1200 °C. Four three-phase regions of the ($\alpha$Co) + ($\epsilon$Co, Re) + $\lambda_3$, $\lambda_3$ + bcc-(Ta) + $\mu$-Co$_6$Ta$_7$, $\lambda_3$ + $\chi$-Re$_7$Ta$_3$ + bcc-(Ta), and ($\epsilon$Co, Re) + $\lambda_3$ + $\chi$-Re$_7$Ta$_3$ were experimentally determined. The 1200 °C isothermal section was similar to the 1100 °C isothermal section. The solubility of Re in the $\lambda_3$ phase increased a little from 20.1 at. % at 1100 °C to 21.5 at. % at 1200 °C. The solubility of Re in the $\lambda_2$ phase was about 3.5 at. %, almost the same as at 1100 °C. The $\mu$-Co$_6$Ta$_7$ and CoTa$_2$ phases dissolved about 3.8 at. % and 1.4 at. % Re, respectively. The solubility of Ta in the ($\epsilon$Co, Re) phase was about 24.1 at. %. The solubility of Co in the $\chi$-Re$_7$Ta$_3$ phase was measured to decrease from 16.8 at. % to 13.4 at. %. The three-phase region of $\lambda_3$ + $\chi$-Re$_7$Ta$_3$ + bcc-(Ta) at the 1200 °C isothermal section was larger than that of the 1100 °C isothermal section, while the ($\epsilon$Co, Re) + $\lambda_3$ + $\chi$-Re$_7$Ta$_3$ three-phase region was smaller.

The isothermal section at 1300 °C is shown in Figure 5c. Compared to the 1200 and 1300 °C isothermal sections, another Laves phase of $\lambda_1$, which existed at 1293 °C~1587 °C in the Co-Ta binary system, appeared in the 1100 °C isothermal section. However, the solubility of Re was so small that it was not identified. The 1300 °C isothermal section had one more three-phase region ($\lambda_1$ + $\lambda_2$ + $\mu$-Co$_6$Ta$_7$) than in the 1100 and 1200 °C isothermal sections. The solubility of Re in the $\lambda_3$ and $\lambda_2$ phases was about 22.3 at. %, 3.6 at. %, respectively. The solubility of Re in $\mu$-Co$_6$Ta$_7$ was found to be about 6.1 at. %, occupying the most out of all the investigated isothermal sections. The CoTa$_2$ phase dissolved about 1.8 at. % Re. The solubility of Ta in the ($\epsilon$Co, Re) phase was about 23.2 at. %. There were seven three-phase regions of ($\alpha$Co) + ($\epsilon$Co, Re) + $\lambda_3$, $\lambda_3$ + bcc-(Ta) + $\mu$-Co$_6$Ta$_7$, $\lambda_3$ + $\chi$-Re$_7$Ta$_3$ + bcc-(Ta), ($\epsilon$Co, Re) + $\lambda_3$ + $\chi$-Re$_7$Ta$_3$, $\lambda_3$ + $\lambda_2$ + $\mu$-Co$_6$Ta$_7$, $\lambda_1$ + $\lambda_2$ + $\mu$-Co$_6$Ta$_7$, and CoTa$_2$ + $\mu$-Co$_6$Ta$_7$ + bcc-(Ta) existing at 1300 °C. The former four three-phase regions were experimentally confirmed, and the last three three-phase regions were not experimentally evidenced. The three-phase equilibrium of $\lambda_3$ + bcc-(Ta) + $\mu$-Co$_6$Ta$_7$ was smaller than those in the 1100 and 1200 °C isothermal sections.

## 4. Conclusions

The isothermal sections of the Co-Re-Ta ternary system at 1100, 1200, and 1300 °C were experimentally investigated. The results were as follows: (1) There were six three-phase regions at the 1100 and 1200 °C isothermal sections and seven three-phase regions at the 1300 °C isothermal section; (2) the ($\varepsilon$Co, Re) phase had a large solubility of Ta and extended from the Re-rich side to the Co-rich side; (3) the $\lambda_3$ phase, with a large solubility of Re, surrounded the $\lambda_2$ phase which dissolved a little Re; (4) the solubility of Re in the CoTa$_2$ phase changed little while the solubility of Re in the $\mu$-Co$_6$Ta$_7$ phase increased with the temperature increase from 1100 to 1300 °C; (5) no ternary compound was found.

**Author Contributions:** Conceptualization, X.L. and C.W.; funding acquisition, X.L. and C.W.; investigation, D.W., J.Z. (Jinbin Zhang), J.H. and Y.L.; supervision, X.L. and C.W.; writing—original draft, D.W.; writing—review and editing, M.Y., J.Z. (Jiahua Zhu), L.L., Y.C. and S.Y.

**Funding:** This work was supported by the National Key R&D Program of China (Grant No. 2016YFB0701401) and National Natural Science Foundation of China (Grant Nos. 51831007 and 51471138).

**Acknowledgments:** This work was supported by the National Key R&D Program of China (Grant No. 2016YFB0701401) and National Natural Science Foundation of China (Grant Nos. 51831007 and 51471138).

**Conflicts of Interest:** The authors declare no conflicts of interest.

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
