# Peer review of "Experimental Investigation of Phase Equilibria in the Co-Re-Ta Ternary System"

_metals, doi:10.3390/met8110911_

Round 1
Reviewer 1 Report
Authors present quite comprechensive study of refractory ternary system with industrial relevance. Authors give experimental evidences for phase equilibria in the ternary system based on BSE and PXRD. The manuscript might potentially be visible in the community and has broad interest for metallurgists. Nevertheless, I recomend to improve the manuscript before considering for the publication.
Abstract and Introduction should be written more cerefully, minor grammar polishing should be done. In the introduction authors discuss the relevance of Re for alloying, but do not discuss well-established "RHENIUM EFFECT", I suggest to give some references as well as to discuss the current status of Re-effect in the context of Co/Ta alloys.
Authors refere to recent works on Re-Ta, Re-Co and Ta-Co binaries, nevertheless, all sstems were investigated in many details in late 60-th and early 70-th. I recommend to give all relevant references and critically assess all binaries to have all information in one single place. It is not very clear if there is any previous information about Co-Re-Ta ternary in the literature.
Authors refere to thermodynamic assessment of binaries. I would be helpful to have all three binary thermodynamic databases and a section with relevant discussion of CALPHAD or similar modelling of the systems. I also recommend to invest time into thermodynamic modelling of the ternary system, which might help to understand phase diagtam better.
Mechanical properties and corrosion resistance of alloys are the most important information for the community. Nevertheless, authors do not give any properties of discussed system. I suggest to invest time into mechanical or/end electrochemical characterisation of alloys (authors in any case have all samples and could do it easily). Such information would make the manuscript more complete and relevant.
Author Response
Response to Reviewer 1 Comments
Metals
Manuscript ID: metals-383571
Title: Experimental investigation of phase equilibria in the Co-Re-Ta ternary system by Xingjun Liu, Dan Wu, Jinbin Zhang, Mujin Yang, Jiahua Zhu, Lingling Li, Yuechao Chen, Shuiyuan Yang, Jiajia Han, Yong Lu and Cuiping Wang.
Dear Reviewer:
Thank you very much for your critical comments and kind suggestions on our manuscript entitled “Experimental investigation of phase equilibria in the Co-Re-Ta ternary system”(Manuscript ID: metals-383571). We have considered your comments and suggestions over and over again. And then the manuscript has been revised according to these comments. All the modifications have been highlighted in the revised version.
Point 1: Abstract and Introduction should be written more carefully, minor grammar polishing should be done. In the introduction authors discuss the relevance of Re for alloying, but do not discuss well-established "RHENIUM EFFECT", I suggest to give some references as well as to discuss the current status of Re-effect in the context of Co/Ta alloys.
Response 1:
Thank you for your kind suggestion. We are sorry for the mistake of the word “solubilitiies” in line 14 and have changed it into “solubilities”. In terms of "RHENIUM EFFECT", we have discussed it in the introduction which is highlighted.
A detailed explanation of change:
“Re is an important solid-solution strengthening element which can retard γ’ phase coarsening and enhance the creep-rupture property [2,3].” changed to “Re does not randomly distribute in the alloy, it hinders dislocation movement by forming tiny clusters which acts as obstacles during creep tests [2-5]. Thus, the addition of Re can effectively enhance the creep properties of superalloys. Meanwhile, as Re element increases, it will fine the morphology and enhance the content of the alloy compound for co-based superalloys [6].”
Point 2: Authors refer to recent works on Re-Ta, Re-Co and Ta-Co binaries, nevertheless, all systems were investigated in many details in late 60-th and early 70-th. I recommend to give all relevant references and critically assess all binaries to have all information in one single place. It is not very clear if there is any previous information about Co-Re-Ta ternary in the literature.
Response 2:
Many thanks for the careful reading and your kind comment sincerely. According to your suggestion, the relevant information for the three binary systems has been discussed in the introduction, respectively. In addition, we have searched literature and found that there is no information about the experimental investigation and thermodynamic data of Co-Re-Ta ternary system. It has been made clear in the paper now.
A detailed explanation of change:
For Co-Re system, the text “Elliott [12] published the results about the Co-Re binary system. Later, Predel [13] reported the Co-Re binary system based on experimental data. Recently, Liu et al. [14] and Guo et al. [15] estimated the Co-Re system and it was consistent with the experimental data. The Co-Re phase diagram newly assessed by Guo et al. [15] is applied in this work.”has been added in the introduction.
For Re-Ta system, the text “Greenfield and Beck [16] firstly studied the Re-Ta binary system, they investigated alloys with Ta contents between 25 and 52 at% and reported the composition range of σ and χ phase. Cui and Jin [17] treated the σ phase as a stoichiometric phase and thermodynamically assessed the Re-Ta system. Afterwards, Liu and Chang [18] also evaluated the thermodynamic description of Re-Ta system. Recently, Guo et al. [15] estimated Re-Ta phase diagram with the latest thermodynamic description for pure Re and this Re-Ta binary system is used in the paper.” has been added in the introduction.
For Co-Ta system, the text “The Co-Ta binary system has been investigated by many researchers [19-24]. Itoh et al. [19] investigated the homogeneity ranges, crystal structures and magnetic properties of the three Laves phases in the Co-Ta system. Okamoto [20] assessed Co-Ta phase diagram and treated C14 as line compound. Liu and Chang [21] thermodynamically assessed the Co-Ta binary system.” has been added in the introduction.
The text “However, the experimental phase diagram of the Co-Re-Ta ternary system have not been reported yet.” changed to “However, there is no information about the experimental investigation and thermodynamic data of the Co-Re-Ta ternary system.”
Point 3: Authors refer to thermodynamic assessment of binaries. I would be helpful to have all three binary thermodynamic databases and a section with relevant discussion of CALPHAD or similar modelling of the systems. I also recommend to invest time into thermodynamic modelling of the ternary system, which might help to understand phase diagram better.
Response 3:
Thank you for your nice suggestion. Our experimental results are useful for the supplement of thermodynamic database and we are working on the thermodynamic modeling of the Co-Re-Ta ternary system. However, the work is not yet mature. We hope that it could be completed and published in Metals as soon as possible.
Point 4: Mechanical properties and corrosion resistance of alloys are the most important information for the community. Nevertheless, authors do not give any properties of discussed system. I suggest to invest time into mechanical or/end electrochemical characterization of alloys (authors in any case have all samples and could do it easily). Such information would make the manuscript more complete and relevant.
Response 4:
Thank you for your helpful suggestion. To be honest, we think mechanical properties and corrosion resistance of alloys are the most important information for the community too. For example, based on the phase equilibria result of the (αCo)+λ3 two-phase region, some Laves phase (λ3) strengthened Co-based alloys were designed. We are working on its mechanical properties now. We also hope that all the relevant results could be published in Metals in the future.
Thank you for your comments and suggestions again.
Yours sincerely,
Prof. Cuiping Wang (Corresponding author)
Department of Materials Science and Engineering
College of Materials
Xiamen University, Xiamen 361005, P. R. China
Tel: +86-592-2180606
Fax: +86-592-2187966
E-mail: wangcp@xmu.edu.cn

Reviewer 2 Report
Dear authors,
This is a very straight forward work. You did a good experimental job clearly described and structured. The experimental results are also clear and well documented. My comment regarding the introduction is that you did not so clearly explained the scope of the selection of the experimentally investigated phase equilibria of the Co-Re-Ta ternary system at 1100, 1200 and 1300°C respectively. Thus, please introduce in the text a short note with the scope of the selection.
Figure 1 is very nice. Please consider to explain to the reader how to must read this graph with respect to the selected temperatures.
With regard to section 3, please provide some additional explanations regarding the selection of the annealing duration ( time in days varied from 15 to 650). How did you end up to these numbers?
Some minor things:
· In line 64, in table 1 you have a German word “Strukturbericht”. Is that on purpose? I don’t think so. Thus, please correct it.
· In line 66 you use capital letter for Rhenium, but plain for tantalum and cobalt. Please improve accordingly in order to have a uniform presentation style.
English and grammar are fine. A minor spell check is always recommended.
Just out of curiosity, did you perform any micro-hardness testing in the investigated phases and microstructures? If yes, surely this is of interest and could enhance section 3 (results & discussion).
Herewith I accept the publication of your manuscript after a minor revision in order to enhance the text in the aforementioned points.
Best regards,
The reviewer
Author Response
Response to Reviewer 2 Comments
Metals
Manuscript ID: metals-383571
Title: Experimental investigation of phase equilibria in the Co-Re-Ta ternary system by Xingjun Liu, Dan Wu, Jinbin Zhang, Mujin Yang, Jiahua Zhu, Lingling Li, Yuechao Chen, Shuiyuan Yang, Jiajia Han, Yong Lu and Cuiping Wang.
Dear Reviewer:
Thank you very much for your critical comments and kind suggestions on our manuscript entitled “Experimental investigation of phase equilibria in the Co-Re-Ta ternary system”(Manuscript ID: metals-383571). We have considered your comments and suggestions over and over again. And then the manuscript has been revised according to these comments. All the modifications have been highlighted in the revised version.
Point 1: You did not so clearly explained the scope of the selection of the experimentally investigated phase equilibria of the Co-Re-Ta ternary system at 1100, 1200 and 1300°C respectively. Thus, please introduce in the text a short note with the scope of the selection.
Response 1:
Thank you for your nice question. We selected the scope of temperature from 1100 to 1300°C for the two reasons: (1) the Co-based superalloys are widely used in high-temperature areas such as aircraft engine and turbine blades. It is more meaningful to investigate the phase equilibria at high temperature. (2) Our laboratory mainly conducts the work of high-temperature alloys and there is a eutectoid reaction of λ1λ2 + μ-Co6Ta7 at 1293°C. Therefore, the scope of temperature from 1100 to 1300°C is selected to investigate the phase equilibria of the Co-Re-Ta ternary system.
A short note “The temperature of 1100, 1200 and 1300°C is selected because the co-based superalloys are widely used in high-temperature area such as aircraft engine and turbine blades. It is more meaningful to investigate the phase equilibria at high temperature.” has been added and highlighted in the paper to explain the scope of selection.
Point 2: Figure 1 is very nice. Please consider to explain to the reader how to must read this graph with respect to the selected temperatures.
Response 2:
Thanks for your careful reading. Firstly, the Co-Re and Re-Ta binary systems are relatively simple, the temperature is mainly selected according to the Co-Ta binary system. Secondly, the Co-based superalloys are widely used in the high-temperature area. Hence, we choose the high temperature to conduct our work. Thirdly, between 1200 and 1300°C, there is a eutectoid reaction of λ1 λ2 + μ-Co6Ta7 existing in the Co-Ta binary system. Therefore, 1100, 1200 and 1300°C are selected to have a more comprehensive investigation on Co-Re-Ta ternary system. According to above points, the temperature of 1100, 1200 and 1300°C is selected to investigate the phase equilibria of the Co-Re-Ta ternary system.
Point 3: With regard to section 3, please provide some additional explanations regarding the selection of the annealing duration ( time in days varied from 15 to 65). How did you end up to these numbers?
Response 3:
Thank you for your nice question. We select the annealing duration for the several reasons:
(1) the difference of the temperature and element contents. Generally, the higher the annealing temperature is, the shorter annealing duration is. In addition, at the same temperature, if the content of Re is higher, the annealing time is longer.
(2) Also, at first, we select the annealing time according to the similar experimental ternary system and the experiment experience of our laboratory.
(3) After annealing, we checked the microstructure to judge if the annealing time is enough. If the alloy reached phase equilibria, the phase boundaries should be distinct and clear. If not, the annealing treatment time will be increased next time.
Thanks again.
Point 4: In line 64, in table 1 you have a German word “Strukturbericht”. Is that on purpose? I don’t think so. Thus, please correct it. In line 66 you use capital letter for Rhenium, but plain for tantalum and cobalt. Please improve accordingly in order to have a uniform presentation style.
Response 4:
Thank you for your careful review and critical question. We apologize for the careless mistake. The word “Strukturbericht” in table 1 has been revised for “Structure type”. Also, the word “Rhenium” in line 82 has been changed to “rhenium”. Thanks again.
Point 5: did you perform any micro-hardness testing in the investigated phases and microstructures? If yes, surely this is of interest and could enhance section 3 (results & discussion).
Response 5:
I am sorry that we did not perform any micro-hardness testing in the investigated phases and microstructures. The aim of this paper is to investigate the phase equilibria of Co-Re-Ta ternary system at 1100, 1200 and 1300°C to provide theoretical basis for alloy design and thermodynamic calculation. Instead, based on the phase equilibria result, some Laves phase (λ3) strengthened Co-based alloys were designed and the mechanical tests are ongoing now. All the relevant results will be considered to publish in Metals later. Thanks again.
Thank you for your comments and suggestions again.
Yours sincerely,
Prof. Cuiping Wang (Corresponding author)
Department of Materials Science and Engineering
College of Materials
Xiamen University, Xiamen 361005, P. R. China
Tel: +86-592-2180606
Fax: +86-592-2187966
E-mail: wangcp@xmu.edu.cn

Round 2
Reviewer 1 Report
authors address my comments.